# Evaluating Different Quantitative Shear Wave Parameters of Ultrasound Elastography in the Diagnosis of Lymph Node Malignancies: A Systematic Review and Meta-Analysis

**DOI:** 10.3390/cancers14225568

**Published:** 2022-11-13

**Authors:** Yujia Gao, Yi Zhao, Sunyoung Choi, Anjalee Chaurasia, Hao Ding, Athar Haroon, Simon Wan, Sola Adeleke

**Affiliations:** 1UCL Medical School, University College London, London WC1E 6BT, UK; 2Imperial College London, School of Medicine, London SW7 2DD, UK; 3GKT School of Medicine, King’s College London, London WC2R 2LS, UK; 4Department of Nuclear Medicine, Barts Health NHS Trust, London E1 1BB, UK; 5Institute of Nuclear Medicine, University College London, London NW1 2BU, UK; 6Department of Oncology, Guy’s and St Thomas’ NHS Foundation Trust, London SE1 9RT, UK; 7School of Cancer & Pharmaceutical Sciences, King’s College London, London WC2R 2LS, UK

**Keywords:** elastography, shear wave elastography (SWE), lymph node malignancy, lymphoma, ultrasound elastography

## Abstract

**Simple Summary:**

Ultrasound (US) imaging is a safe, convenient imaging method for identifying malignant lymph nodes. Shear wave elastography (SWE), as a type of US elastography offers the mechanical information of tissue by sensing shear wave propagation in lymph nodes. Malignant lymph nodes can show increased stiffness at the lesion margin and adjacent tissue on the SWE image. However, the diagnostic accuracies of various SWE parameters that quantify tissue stiffness, such as maximum stiffness, mean stiffness, minimum stiffness, and standard deviation, are yet to be demonstrated. We included sixteen eligible studies to evaluate the pooled diagnostic accuracy of different SWE parameters. SWE has demonstrated promise as an imaging modality in diagnosing and differentiating malignancy from benign lymph nodes. Its incorporation into standard US allows for a better evaluation of the target region or lymph node and might reduce the need for invasive procedures or exposure to ionising radiation without compromising on diagnostic accuracy.

**Abstract:**

Shear wave elastography (SWE) has shown promise in distinguishing lymph node malignancies. However, the diagnostic accuracies of various SWE parameters that quantify tissue stiffness are yet to be demonstrated. To evaluate the pooled diagnostic accuracy of different SWE parameters for differentiating lymph node malignancies, we conducted a systematic screening of four databases using the PRISMA guidelines. Lymph node biopsy was adopted as the reference standard. E_max_ (maximum stiffness), E_mean_ (mean stiffness), E_min_ (minimum stiffness), and E_sd_ (standard deviation) SWE parameters were subjected to separate meta-analyses. A sub-group analysis comparing the use of E_max_ in cervical (including thyroid) and axillary lymph node malignancies was also conducted. Sixteen studies were included in this meta-analysis. E_max_ and E_sd_ demonstrated the highest pooled sensitivity (0.78 (95% CI: 0.69–0.87); 0.78 (95% CI: 0.68–0.87)), while E_mean_ demonstrated the highest pooled specificity (0.93 (95% CI: 0.88–0.98)). From the sub-group analysis, the diagnostic performance did not differ significantly in cervical and axillary LN malignancies. In conclusion, SWE is a promising adjunct imaging technique to conventional ultrasonography in the diagnosis of lymph node malignancy. SWE parameters of E_max_ and E_sd_ have been identified as better choices of parameters for screening clinical purposes.

## 1. Introduction

Lymph node (LN) analysis is commonly used for assessing cancer recurrences, metastases, and pre-treatment cancer TNM staging [1]. The abnormal change in LN size, consistency and number is associated with either benign or malignant pathological phenomena [2,3,4]. For example, painless hardened LNs are frequently indicators of metastatic cancer or granulomatous diseases, whereas firm and rubbery LNs may indicate lymphoma [5]. Nodal status assessment is essential in patients with known cancer malignancies as it helps tailor treatment options, decision-making, and predicting prognosis [6]. Therefore, the accurate identification of the nature of lymphadenopathy is crucial yet clinically difficult due to the problems in differentiating malignant lesions from benign ones [7].

Ultrasound (US) imaging is a common complementary imaging modality for the evaluation of lymph nodes due to its safety, convenience, and relative low costs [8]. Sonographic features, such as cystic transformation, microcalcification, hyperechogenicity, or aberrant vascularity, have been linked to LN malignancy. However, no single US reporting standard for identifying malignant LNs with satisfactory sensitivity and specificity has been developed [9,10]. 

The newly emerged technique, US elastography, first proposed by Ophir et al [4], offers mechanical information of tissue by sensing internal tissue deformation or displacement in response to an applied force and displays the resulting information in a suitable form [11]. Since malignant LNs are frequently infiltrated by tumour cells, tumoral necrosis, and calcification but benign LNs are not, the stiffness difference between malignant and benign LNs can provide important clinical information to differentiate LN states [12]. Similar to standard ultrasound imaging, US elastography overcomes the limitations of manual examination, for example limited accessibility to organs, subjective interpretation, and inaccurate disease localization [13,14]. In addition, it provides information on lesions that may not have sonographic contrast that otherwise could have been detected ultrasonically [15]. Several studies have documented the utility of US elastography in differentiating malignant from benign lesions in breast, thyroid, prostate, and liver, as well as lymph nodes [8,16,17,18,19,20], and encouraging findings with higher diagnostic accuracy than B-mode in standard ultrasound [21,22] or as an adjunct technique have been observed [16,23]. 

Depending on the type of force applied, either quasi-static (probe palpation or compression) or dynamic (thumping or vibrating), US elastography could examine the tissue deformation in two ways: strain elastography (SE), which measures tissue strain, or shear wave elastography (SWE) measuring shear wave propagation [11,24]. The current literature associates SWE to methods visualising shear-wave speed using acoustic radiation force [11], including both two-dimensional and three-dimensional SWE. Transverse orientated shear waves are generated and propagated through target tissues with varying elasticities at different velocities [25]. The final elasticity can be directly and qualitatively interpreted via a colour map overlaid on the B-mode ultrasonic image or through conversion into Young’s modulus, in kilopascals (kPa), for quantitative analysis within a region of interest [18,26,27]. The qualitative analysis of SWE classifies elastograms into various colour patterns [28]; however, the number of patterns comparably vary between study groups hampering the results. Malignant lymph nodes can show the stiff rim sign: increased stiffness at the lesion margin and adjacent tissue on the elastogram. The region of interest (ROI) is set over the stiffest part of the immediate adjacent stiff tissue, and the abnormal lymph nodes can be quantified using shear wave values [29]. 

The current literature reports different quantitative SWE parameters: E_min_ (minimum stiffness), E_mean_ (mean stiffness), E_max_ (maximum stiffness), and E_sd_ (one standard deviation of stiffness) with different cut-off values for different anatomical lymph node positions [12,16,19,22,29,30,31,32,33,34,35]. However, more evidence is needed to discuss the significance of the different SWE parameters and their diagnostic potential for lymph node diseases. The aim of this study is to investigate the role of SWE parameters as potential predictive parameters in lymph node malignancies and to evaluate their diagnostic accuracy for the differentiation between benign and malignant lymph nodes.

## 2. Materials and Methods

### 2.1. Literature Search

A systematic search was carried out using Medline, Ovid, PubMed, Embase, and Web of Science databases to retrieve all studies that contribute relevant evidence from inception to date. The search included Medical Subject Heading (MeSH) terms and free text with appropriate Boolean operators. The following terms were included in the search: “shear wave elastography”, “elastography”, “lymph node OR lymph gland”, and “diagnosis”, as well as multiple synonyms, were used to account for differences in terminology. In the case of absent or ambiguous data, the corresponding authors were contacted directly for clarification.

Eligible studies were screened independently by two researchers (Y. Gao and Y. Zhao). Articles considered eligible had the full text in English retrieved for further review. Any disagreements between reviewers were discussed until a consensus was reached, and in cases where this was not possible, a third case reviewer was consulted. The study was designed using the PRISMA-P (Preferred Reporting Items for Systematic Reviews and Meta-analysis Protocols) and was registered with PROSPERO (CRD42022322053). All exclusions were noted for further analysis and the reasons were documented in detail to generate the PRISMA flow (Figure 1 ) [36]. We continued updating the literature search until August 2022 and the bibliographies of articles were also screened to identify all appropriate articles. 

Search terms used are: (((((elastography OR strain elastography OR real time elastography OR shear wave elastography) AND (lymph node OR lymph gland OR disease)) AND Meta*) AND Diagno*) AND Accurac*). ti,ab.

### 2.2. Data Collection 

All desired data were collected and summarised in a dedicated datasheet (Table 1). Study characteristics, such as authors, year of publication and study design, and the demographics of patient population, patient mean age, number of lymph node lesions, and the lymph node type, were noted. Information from study methodology and outcomes, including SWE imaging strength, SWE parameters/settings, elasticity cut-off values, and mean elasticity value in lymph node malignancy, as well as histopathology methods and definition for clinically significant disease, were reported. 

### 2.3. Inclusion Criteria

Studies which included patients who received shear wave elastography with underlying pathology of lymph node malignancies were considered eligible for inclusion. Studies fulfilling the following requirements were included in the meta-analysis: (1) the study was published in English with accessible full text; (2) shear wave ultrasound elastography was used to assess lymph node status; (3) study design was observational, either prospective or retrospective; (4) the study has used reference standards that was based on histopathology reports, biopsy, or fine needle aspiration; (5) the study contained full set of clinical data that could be directly or indirectly obtained, including true positives, true negatives, false positives, false negatives, sensitivity, specificity, PPV, and NPV to construct diagnostic tables. 

### 2.4. Exclusion Criteria

Studies matching the following criteria were excluded: (1) studies there were on topics other than using the value of elasticity modulus (kPa) to diagnose lymph node malignancies; (2) studies that were not using shear wave elastography but strain elastography; (3) studies that were not performed to differentiate lymph node malignancies; (4) studies that were correspondence articles, expert opinions, conference abstracts, review articles and case reports, or not original papers; and (5) studies with text in a foreign language besides English. 

### 2.5. Quality Assessment 

Study quality was assessed using the Quality Assessment of Accuracy Studies (QUADAS-2) tool [42]. The QUADAS-2 explored the risk of bias in four key domains: patient selection, index test, reference standard and flow and timing. Additionally, concerns regarding the applicability of these domains to the systematic review were investigated.

### 2.6. Endpoints

The primary endpoint was to identify if there was statistically significant evidence in supporting the use of shear wave elastography in lymph node malignancy diagnosis.

### 2.7. Meta-Analysis 

True positives (TPs), true negatives (TNs), false positives (FPs), and false negatives (FNs) were extracted from all selected studies to produce a contingency table based on the histopathology result and SWE parameter used. Pooled quantitative sensitivities and specificities and accuracies were compared using bivariate analysis at a 95% confidence interval (CIs). The summary receiver operating characteristic (SROC) curves were then generated using the area-under-the-curve (AUC) values presented. Heterogeneity and inter-study variation were quantified through I^2^. 

## 3. Results

### 3.1. Study Characteristics

Sixteen studies were considered eligible for further analysis after screening (Table 2) [12,16,19,22,29,30,31,32,33,34,35,37,38,39,40,41]. The studies were published between 2012 to 2022. A total of 1276 patients and 1479 lymph node lesions were included for quantitative SWE parameters analysis. Eleven studies were conducted prospectively [12,16,31,33,34,35,37,38,39,40,41] and five retrospectively [19,22,29,30,32]. Patients’ age ranged from 7 to 86 years old. All studies performed histopathology examination using ultrasound-guided fine needle aspiration biopsy (FNAB), core needle biopsy (CNB), surgically resected specimens or axillary lymph node biopsy (ALNB), or sentinel lymph node biopsy (SLNB) as the reference standard to confirm LN malignancy. The E_max_ parameter was analysed in fifteen studies [12,16,19,22,29,30,31,32,33,34,35,38,39,40,41] eleven studies conducted E_mean_ analysis [12,19,22,29,30,31,33,34,37,39,40], seven studies conducted E_min_ analysis [19,22,29,30,31,34,39], and five studies conducted E_sd_ analysis [22,29,30,39,40]. The cut-off values for individual SWE parameters varied considerably across these studies, ranging between 15.2 to 57 kPa for E_max_ [12,16,19,22,29,30,31,32,33,34,35,38,39,40,41], 15.5 to 30.2 kPa for E_mean_ [12,19,22,29,30,31,33,34,37,39,40], 11.4 to 24 kPa for E_min_ [19,22,29,30,31,34,39], and lastly, 2.1 to 6.9 kPa for E_sd_ [22,29,30,39,40]. The elasticity setting on the kPa display scale was between 0–180 kPa across all studies except Chami et al., which qualitatively displayed between 0–100 kPa, with lowest stiffness at 0 kPa represented by dark blue and highest represented by dark red. 

### 3.2. Risk of Bias

A total of sixteen studies were reviewed, which observed an overall low risk of bias and high applicability to the review question [12,16,19,22,29,30,31,32,33,34,35,37,38,39,40,41] (Table 3). However, four studies were marked as “fair” in terms of overall diagnostic quality. Ng et al. [16] conducted a case-control study and failed to report on the patient identification and recruitment process for the trial. Touresse et al. [33], You et al. [39], and Youk et al. [22] were also classified as “fair”, as the studies did not provide an acceptable description of the flow and timing regarding the index test and standard reference test.

### 3.3. Meta-Analysis 

In order to assess the effect of SWE parameters (E_max_, E_mean_, E_min_, and E_sd_) in diagnosing lymph node diseases, individual meta-analysis was conducted on each diagnostic setting (Figure 2, Figure 3, Figure 4 and Figure 5).

Fifteen studies have investigated the use of E_max_ setting [12,16,19,22,29,30,31,32,33,34,35,38,39,40,41] (Figure 2). The sensitivity was 0.78 (95% CI: 0.69–0.87) and specificity was 0.82 (95% CI: 0.72–0.93) The AUC value was 0.88. Again, high heterogeneities presented for both sensitivity and specificity (*p* < 0.01) across all studies. 

Eleven studies have investigated the use of the E_mean_ setting [12,19,22,29,30,31,33,34,37,39,40] (Figure 3). The sensitivity was 0.67 (95% CI 0.54–0.80) and specificity was 0.93 (95% CI: 0.88–0.98) The AUC value was 0.89. High heterogeneities presented for both sensitivity and specificity (*p* < 0.01).

Seven studies have investigated the use of the E_min_ setting [19,22,29,30,31,34,39] (Figure 4). The sensitivity was 0.60 (95% CI: 0.43–0.76) and specificity was 0.91 (95% CI: 0.85–0.96) The AUC value was 0.87. High heterogeneities presented for both sensitivity and specificity (*p* < 0.01).

Five studies investigated the use of E_sd_ setting [22,29,30,39,40] (Figure 5). The sensitivity was 0.78 (95% CI: 0.68–0.87) and specificity was 0.91 (95% CI: 0.84–0.99). The AUC value was 0.90. The heterogeneity was high for specificity (*p* < 0.01) and slightly lower for sensitivity (*p* = 0.02) but still significant. 

### 3.4. Sub-Group Analysis 

Sub-group analysis was also conducted to compare the use of SWE in cervical and axillary LN malignancies (Figure 6 and Figure 7). Given that E_max_ was the most common SWE parameter reported in LN malignancy detection and that it was hypothesised as the parameter with a higher sensitivity in detecting cortical focal metastases in large, heterogeneous LNs [35,41], as LNs may be partially infiltrated by the tumour or have spatial heterogeneity due to tumoral necrosis [12], it was chosen as the parameter to analyse in this sub-group analysis. 

Five studies performed SWE using E_max_ on axillary lymph nodes [16,22,29,31,33]. The reported pooled sensitivity was 0.74 (95% CI: 0.52–0.95) and specificity 0.86 (95% CI: 0.86–1.00) with an AUC value of 0.9. E_max_ results on cervical lymph nodes, including the thyroid, were available from nine studies [12,19,30,32,34,35,38,39,41]. The pooled sensitivity, specificity, and AUC values were 0.80 (95% CI: 0.69–0.91), 0.80 (95% CI: 0.64–0.96), and 0.87. Heterogeneities remained high (*p* < 0.01) in both sub-groups for both sensitises and specificities. 

## 4. Discussion

This paper is the first systematic review and meta-analysis comparing the diagnostic accuracies of E_min_, E_mean_, E_max_, and E_sd_ SWE parameters in lymph node malignancy. The results show that the E_mean_ parameter had the highest overall specificity of 0.93 (95% CI: 0.88–0.98, I^2^ = 83%) and both E_max_ and E_sd_ had the highest sensitivity of 0.78 (95% CI: 0.69–0.87, I^2^ = 88% and 95% CI: 0.68–0.87, I^2^ = 65%). Five out of the sixteen selected studies had a complete dataset for all four SWE parameters: three indicated E_mean_ as the optimal parameter with highest diagnostic accuracy [22,30,31], while two indicated E_max_ as the optimal [29,39]. However, in our study among all the SWE parameters, E_sd_ had the highest AUC at 0.90, but the AUC values between SWE parameters did not differ significantly. 

All selected studies were conducted in either cervical or axillary lymph nodes, and existing systematic reviews have analysed the use of elastography for both types of lymph node malignancy individually, for example cervical lymphadenopathy by Suh et al. [43] and axillary lymphadenopathy by Huang et al. and Wang et al. [44,45]. However, a mixed choice of SWE parameters was pooled in analysis [43,45], and no comparisons were made between lymph node malignancies in different anatomical positions. This allowed our study to conduct a further sub-group analysis using E_max_ to compare the use of SWE in cervical and axillary LNs and found a higher diagnostic performance in axillary LNs (AUC = 0.90 vs. 0.87). Wang et al. [44] reported a similar sensitivity of 0.73 (95% CI: 0.44–0.91) but a higher specificity at and 0.94 (95% CI: 0.81–0.98) and higher AUC at 0.94 (95% CI: 0.91–0.96) of E_max_ in axillary lymphadenopathy. However, despite a smaller number of included studies (n = 4), their results were likely affected by high specificity value at 89.8 (85.5–94.2) reported from one ex vivo study by Bae et al. [46], introducing bias. 

Studies have [22,31,40] suggested that all four SWE parameters of E_max_, E_mean_, E_min_, and E_sd_ were significantly higher for malignant LNs than benign LNs. Additionally, quantitative SWE features had significantly higher diagnostic performance than conventional grey-scale ultrasound features. Ng et al. [16] reported that the combined use of adjuvant B-mode ultrasound and qualitative SWE provided the optimal accuracy in detecting axillary LNs metastasis in their study sample (sensitivity 71.6%, specificity 95%, AUC = 0.882) in comparison with ultrasound on its own (sensitivity 70.0%, specificity 70.2%, AUC = 0.701). Tourasse et al. and Kim et al. [30,33] both emphasised the importance of high specificity or strong predictive values over false negative concerns, stating that the current main objective of SWE was to facilitate confirmation of metastatic LN diagnosis because high specificity reduces unnecessary biopsies and other invasive or high-risk investigations and enables the selection of LNs that are more likely to be malignant to undergo SNLB or FNA regardless of conventional US findings. In contrast, the high specificity is countered by low sensitivity found by Tourasse et al. ranging from 18% to 36% for E_mean_ cutoffs at 19.0–23.6 kPa and 36% to 45% for E_max_ at 23.7–26.5 kPa. As a result, E_max_ and E_sd_ may be better choices of parameters if the clinical objective is screening, based on higher sensitivity and accuracy performance, and the optimal choice of parameter used will depend on the clinical context when more data become available. 

Studies by Zhao et al. [47] and Tamaki et al. [48] have also reported results of high sensitivity (85.7% and 82.8%) accompanied by relatively low specificity (54.7% and 69.6%) using other available SWE parameter that measured shear wave speed in m/s. The optimal cut-offs of the shear wave speeds differed between groups from 6.42 m/s [47] to 1.44 m/s [48]. Shear wave speed parameters are direct measurements from elastograms and are converted into elasticity SWE parameter based on assumptions, such as constant tissue density, pure tissue elasticity and isotropy, which are generally not valid [11,24]. If we were to include these studies, this conversion would introduce inaccuracy and inconsistency between studies because the linear array transducer was operated at a different bandwidth (4–9 MHz) and SWS measurement are known to be frequency dependent [24], hence they were not included in this meta-analysis. However, future studies should investigate the relationship between the shear wave speed S parameters and elasticity E parameters to allow more accurate conversion and data pooling. Other elasticity SWE parameters, such as E_ratio_ [16,22,39], E_median_ [49], were also encountered in other studies but not widely used. Studies which have carried out comparative analysis between SWE and B-mode ultrasound have shown the beneficial potential of SWE as adjunct to B-mode ultrasound in enhancing the prediction of LN status in several cancers [16,44].

SWE is a novel, non-invasive, non-radioactive imaging modality that has demonstrated greater phantom resolution, fewer stress concentration artefacts related with boundaries and elastic modulus inhomogeneities and enhanced signal-to-noise ratio (SNR) with increasing depth [13]. It is thought to be more operator-independent since shear waves are created by focused high-intensity, short-duration acoustic pulses from an ultrasonic transducer rather than freehand compression as in SE [8,18]. High reproducibility of SWE has been reported by existing data according to intraclass correlation coefficients (ICCs), with the interobserver agreement ranging from 0.66–0.80 [12,50,51] and intra-observer agreement ranging from 0.64 to 0.84 [12,51]. Furthermore, SWE can provide the absolute quantification of tissue stiffness. It represents a considerable advancement over strain elastography, which can only provide the semiquantitative estimates of relative tissue strain [12,24]. Nevertheless, there are some unresolved concerns in US elastography, such as the selection of representative ROIs and unknown manufacturer-related variability in US elastography implementation. There are also a few other practical limitations due to lack of elasticity in cysts and calcified lesions, as well as a focal convex bulge on the skin overlying the ROI, which could produce spuriously stiff elastograms [8], creating scope for future studies to explore these limitations and their impact.

Molecular imaging modalities, such as PET-CT, utilising ^18^F FDG as a biomarker to assess metabolic activity in the LNs have limitations, which can be addressed with developments in SWE technique. Sites with physiological uptake pattern, such as in the testes, it may become impossible to differentiate intense FDG uptake by neoplastic cells from normal physiological testicular activity. The implications are testicular lymphomas are treated with the orchidectomy and combination of chemotherapy. Even after treatment, physiological uptake pattern in contralateral side may be challenging to differentiate from recurrence. Gastric lymphomas may exhibit no or variable metabolic activity and this might have implications in the detection of incidentally detected nodes in the peri-gastric region [52]. Post-treatment LN evaluation is a frequent difficulty encountered in interpretation as differentiating residual nodal uptake with FDG due to lymphoma can be very difficult to differentiate from post-treatment inflammation, co-existing infection, and sites with physiological metabolic activity [53,54]. With further research and advances in this technology, SWE might have clinical impact in the detection of lymph nodes that undergo transformation, resulting in a change in tumour type, as it might be ethically unfeasible to carry out repeated biopsies and repeated cross-sectional imaging with modalities involving the radiation burden in young patients (such as females of child-bearing age and pregnant individuals) or in patients who have contra-indication to MRI. 

This study has some limitations. This study found high heterogeneity ranging from 65% to 94% (Figure 2, Figure 3, Figure 4 and Figure 5) and hence results should be interpreted with caution. An array of reasons for the heterogeneity could be postulated, including study design, measurement methods, reference standard, and characteristics of each study. Five of the twelve studies chosen were retrospective, which may have contributed to selection bias [19,22,29,30,32]. For example, Yang et al. and Seo et al. performed SWE on lymph nodes that were enlarged or abnormal on US findings and thus resulted in a higher proportion of metastatic LNs, contributing to selection bias and sampling error [29,32]. Additionally, direct node-to-node correlation between SWE and surgery was not obtained in all studies [16,22,33]. Tourasse et al. had reported a 10% allowance in nodal size matching between nodes assessed by elastography and nodes surgically excised, which could have resulted in selection errors [33]. With the exception of Choi et al. [35] and You et al. [39], where the averages were 4.5 and 3.6 nodes per patient, respectively, 1.16 lymph nodes per patient were examined on average. Since the number of removed LNs (≥18 in neck dissection) in resected specimens have independent prognostic value, as documented in the pathological literature [54], the low number of LN count per patient in studies selected could manifest itself in sampling error. 

This study highlighted substantial discrepancies in cut-off values ranging from 15.2 to 57 kPa for E_max_, 18.4 to 30.6 kPa for E_mean_, and 11.4 to 24 kPa for E_min_, which likely contributed to the heterogeneity, owing to differences in the reference standards. Different gold standards carry different diagnostic accuracies in determining LN malignancy. For example, studies have suggested that CNB has higher sensitivity than FNAC (87–100% and 74–83.3%, respectively), without significant differences in specificities (96–100% and 98–100%, respectively) [55,56,57]. When compared with surgical biopsy, CNB showed a lower sensitivity at 90% than its specificity at 100% due to false negative cases, and an LN biopsy is recommended in cases of suspected neoplasms [58,59]. In malignancies such as lymphoma, CNB is often preferred over FNAC, mainly to preserve the tissue or nodal architecture to enable comprehensive histopathological review and diagnosis [55,56,57]. In head and neck squamous cell cancers, FNAC are deemed to be adequate in making a diagnosis in certain cases [58,59]. The choice of gold standard needs to be put into context of the malignancy and unique patient circumstances [60]. Though variability in the use of scan technique is the reality of clinical practice, it could make the development or translation of a new imaging technique quite challenging. Surgery, on the other hand, is less prone to the variability of measurements compared to ultrasound; however, it is not always necessary or feasible to carry out nodal dissection in all patients [60,61]. Again, standard clinical practice may influence the way new or upcoming technologies are evaluated and may further compound the variability conundrum.

Depending on the underlying diseases, such as non-Hodgkin’s and Hodgkin’s lymphoma, mantle cell lymphoma, chronic lymphocytic leukaemia, follicular lymphoma, and metastatic malignancy, the elastographic architecture of malignant LNs may be heterogeneous [62,63]. Studies have shown that diffuse and homogenous LN infiltration are more often seen in high-grade lymphomas, whereas focal LN infiltration is indicative of low-grade follicular lymphoma [62]. However, the knowledge of elastography in differentiating grades or types of lymphomas is still very limited. Metastatic LN or lymphoma usually show peripheral or mixed patterns of internal vascularity and are difficult to differentiate, as they could both present as a round heterogenous echoic LN with hilum loss, without definite necrotic area and showing mixed vascularity [63]. It is important to discriminate these entities as the prognosis and treatment differ, and thus advanced imaging techniques are needed to differentiate them for better evaluation and treatment. Future work could explore the elastographic differences between these various types or grades of lymphomas and metastatic LNs by initially analysing their ex vivo characteristics (matched to histology) before the validation of such studies in vivo in human subjects.

Furthermore, since varying ultrasound machines have been employed across the selected studies, the measured values of shear wave speeds may differ between different systems or vendors, in particular, the shear wave vibration and the bandwidth, which may have produced shifts in elasticity values [24], and their impacts on LN diagnosis need to be further investigated in future studies [64]. Differences in software methods used to determine relative shear wave arrival time and speeds may have further led to measurement bias and adjustments for these effects are not attainable [24]. According to the most recent EFSUMB (European Federation of Societies for Ultrasound in Medicine and Biology) guidelines, the use of US elastography has been recommended for targeting malignant LNs for FNA if multiple LNs are present [64]. The difficulty in study comparison due to the lack of standardisation in technique and a definite need to set rigorous cut-off values for each of the various elastography systems and diseases have also been described [24,64]. This could be attributed to a lack of funding [24] and research in this field, emphasising the need for the greater awareness of such technologies. As a result, these findings must be carefully interpreted, with particular attention devoted to heterogeneity issues, and prospective multicentre population-based clinical trials are required to demonstrate the diagnostic value of SWE for lymph node malignancy while assuring consistency across studies to help determine whether results are acceptable clinically.

## 5. Conclusions

In conclusion, SWE has demonstrated promise as an imaging modality in diagnosing and differentiating malignancy from benign lymph nodes. Its incorporation into standard US not only allows for a better evaluation of the target region or lymph node but might reduce the need for invasive procedures or exposure to ionising radiation without compromising on diagnostic accuracy. 

## Figures and Tables

**Figure 1 cancers-14-05568-f001:**
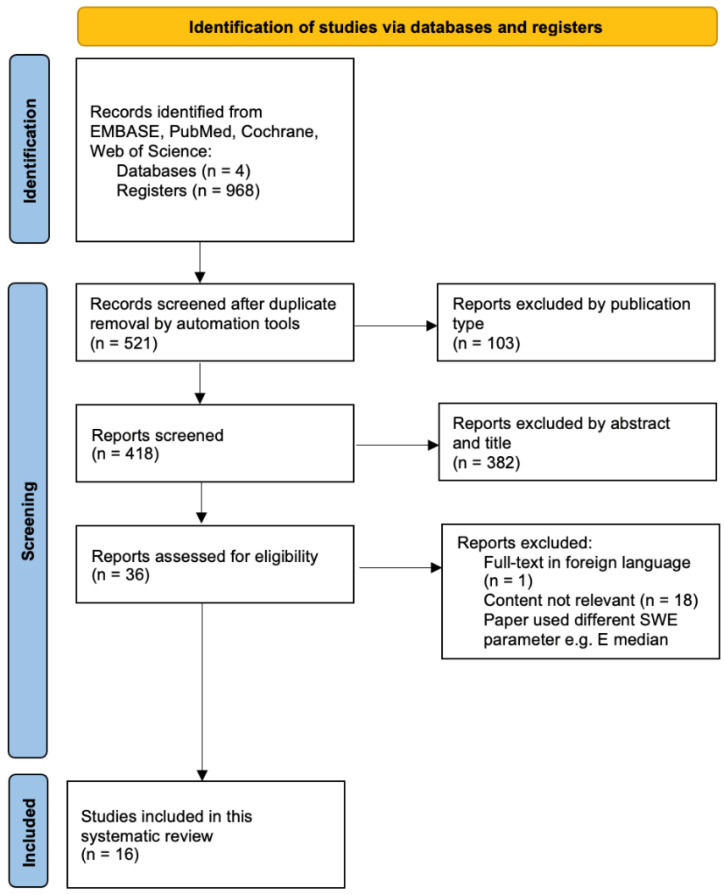
PRISMA flow diagram of evidence acquisition.

**Figure 2 cancers-14-05568-f002:**
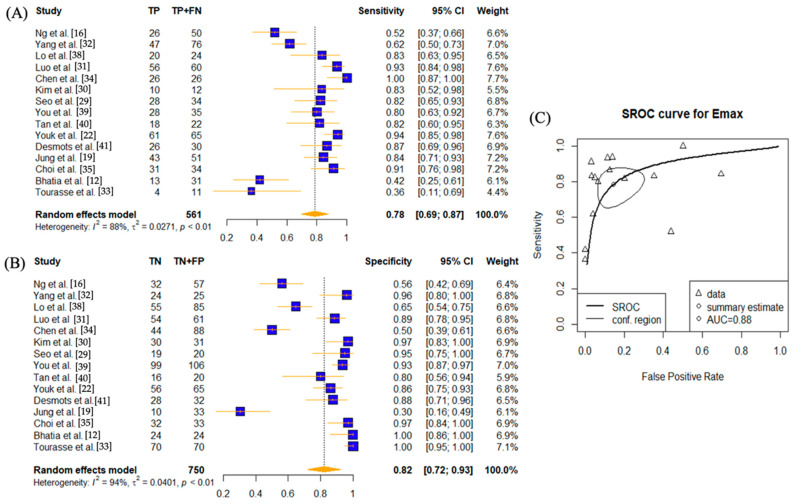
Sensitivity and specificity forest plots and overall accuracy results of E_max_. (**A**–**C**) Reported sensitivity and specificity values for E_max_ values per selected study with AUC value on SROC curve for analysis of E_max_. Forest plots for pooled sensitivities and specificities are displayed as diamonds in the graphs for E_max_ (**A**,**B**). The SROC curve indicates the summary estimates in circles ((**C**) for E_max_). Triangles represent included study with dotted lines representing the confidence interval and solid line for the SROC. AUC value is displayed (in the legend). E_max_—maximum tissue stiffness SWE parameter; AUC—area under the curve; SROC—summary receiver operating characteristic.

**Figure 3 cancers-14-05568-f003:**
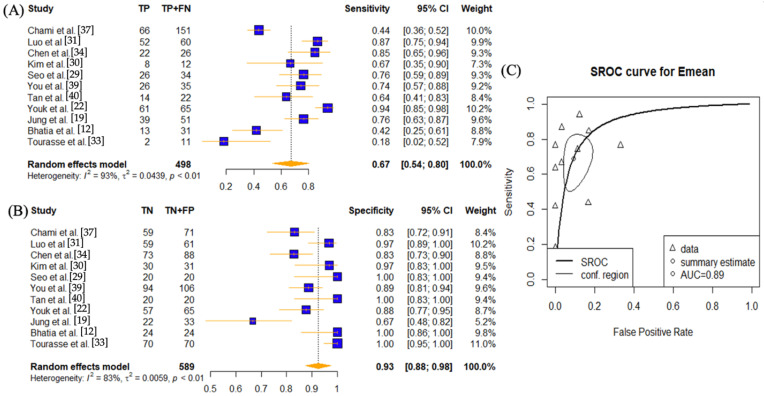
Sensitivity and specificity forest plots and overall accuracy results of E_mean_. ((**A**–**C**) Reported sensitivity and specificity values for E_mean_ values per selected study with AUC value on SROC curve for the analysis of E_mean_. Forest plots for pooled sensitivities and specificities are displayed as diamonds in the graphs for E_mean_ (**A**,**B**). The SROC curve indicates the summary estimates in circles ((**C**) for E_mean_). Triangles represent included study with dotted lines representing the confidence interval and solid line for the SROC. AUC value is displayed (in the legend). E_mean_—average tissue stiffness SWE parameter; AUC—area under the curve; SROC—summary receiver operating characteristic.

**Figure 4 cancers-14-05568-f004:**
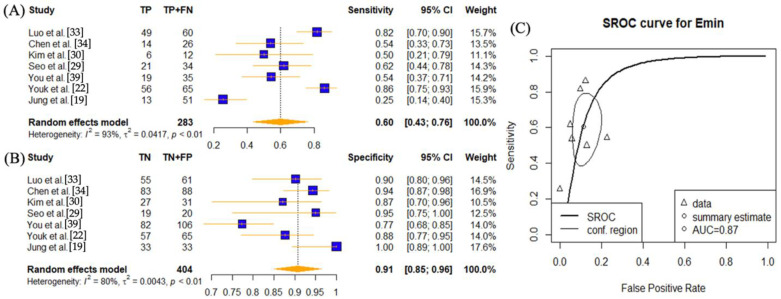
Sensitivity and specificity forest plots and overall accuracy results of E_min_. ((**A**–**C**) Reported sensitivity and specificity values for E_min_ values per selected study with AUC value on SROC curve for the analysis of E_min_. Forest plots for pooled sensitivities and specificities are displayed as diamonds in the graphs for E_min_ (**A**,**B**). The SROC curve indicates the summary estimates in circles ((**C**) for E_min_). Triangles represent included study with dotted lines representing the confidence interval and solid line for the SROC. AUC value is displayed (in the legend). E_min_—minimum tissue stiffness SWE parameter; AUC—area under the curve; SROC—summary receiver operating characteristic.

**Figure 5 cancers-14-05568-f005:**
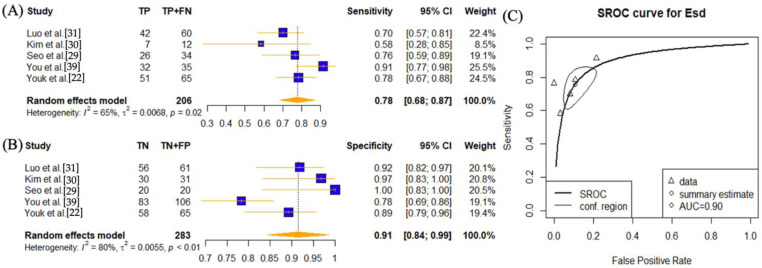
Sensitivity and specificity forest plots and overall accuracy results of E_sd_. ((**A**–**C**) Reported sensitivity and specificity values for E_sd_ values per selected study with AUC value on SROC curve for the analysis of E_sd_. Forest plots for pooled sensitivities and specificities are displayed as diamonds in the graphs for E_sd_ (**A**,**B**). The SROC curve indicates the summary estimates in circles ((**C**) for E_sd_). Triangles represent included study with dotted lines representing the confidence interval and solid line for the SROC. AUC value is displayed (in the legend). E_sd_—tissue stiffness standard deviation SWE parameter; AUC—area under the curve; SROC—summary receiver operating characteristic.

**Figure 6 cancers-14-05568-f006:**
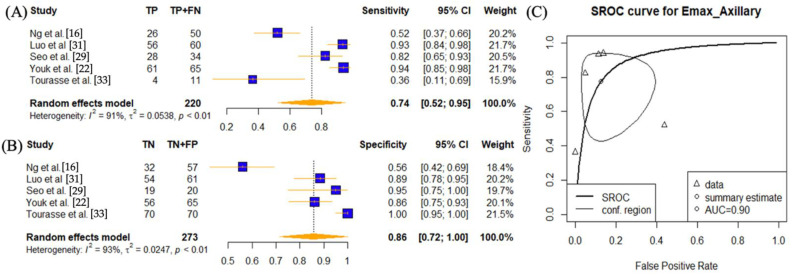
Sensitivity and specificity forest plots and overall accuracy results of E_max_ in Axillary LNs. ((**A**–**C**) Reported sensitivity and specificity values for E_max_ values per selected study with AUC value on SROC curve for the analysis of E_max_. Forest plots for pooled sensitivities and specificities are displayed as diamonds in the graphs for E_max_ (**A**,**B**). The SROC curve indicates the summary estimates in circles ((**C**) for E_max_). Triangles represent included study with dotted lines representing the confidence interval and solid line for the SROC. AUC value is displayed (in the legend). E_max_—maximum tissue stiffness SWE parameter; AUC—area under the curve; SROC—summary receiver operating characteristic.

**Figure 7 cancers-14-05568-f007:**
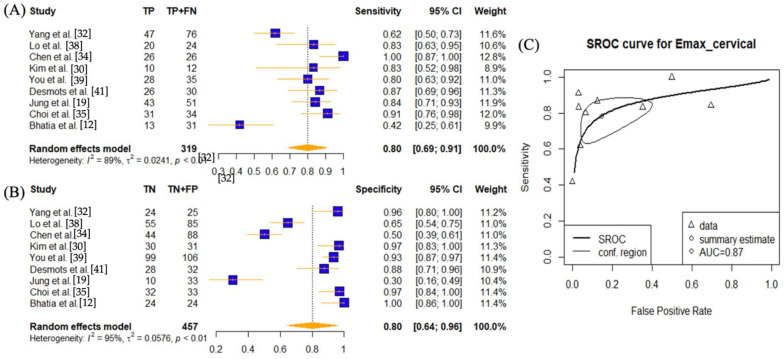
Sensitivity and specificity forest plots and overall accuracy results of E_max_ in cervical (including thyroid) LNs. ((**A**–**C**) Reported sensitivity and specificity values for E_max_ values per selected study with AUC value on SROC curve for the analysis of E_max_. Forest plots for pooled sensitivities and specificities are displayed as diamonds in the graphs for E_max_ (**A**,**B**). The SROC curve indicates the summary estimates in circles ((**C**) for E_max_). Triangles represent included study with dotted lines representing the confidence interval and solid line for the SROC. AUC value is displayed (in the legend). E_max_—maximum tissue stiffness SWE parameter; AUC—area under the curve; SROC—summary receiver operating characteristic.

**Table 1 cancers-14-05568-t001:** Study characteristics of included studies.

Authors	Year of Publication	Study Design	No. of Patients	Mean Age (Range) (Years Old)	No. of Lymph Node Lesions	US Imaging System	SWE Imaging Strength (MHz)	SWE Parameters	Elastic Modulus Values in Malignant Lymph Nodes Mean ± SD or Median (IQR) (kPa)	LN Type	Reference Standard	Clinically Significant Definition
Ng et al. [16]	2022	Prospective	107	58 (32–82)	107	Aixplorer (SuperSonic Imagine)	15-4	E_max_	40.0 ± 46.4	Axillary	ALND or SLNB	Bloom and Richardson Grading ALN pathological staging: cut-off point of >3 mm
								E_mean_	28.9 ± 36.4			
								E_min_	22.5 ± 33.5			
								E_sd_	8.7 ± 27.1			
Chami et al. [37]	2021	Prospective	222	N/A	222	Aixplorer (SuperSonic Imagine Ltd, Aix-en- Provence, France)	SL10-2 (central frequency at 10 MHz)	E_max_	36.1 ± 33.7 (lymphoma) 62 ± 58.2 (carcinoma)	Axillary, Head & Neck, Inguinal	CNB	Doppler criteria
								E_mean_	16.7 ± 12.3 (lymphoma) 29.5 ± 32.3 (carcinoma)			
								E_sd_	6.4 ± 5.7 (lymphoma) 11.1 ± 10.6 (carcinoma)			
Yang et al. [32]	2021	Retrospective	103	43.9 (18–66)	109	Aixplorer (SuperSonic Imagine, Aixen-Provence, France)	15-4	E_max_	34.2 ± 7.0	Cervical	US-guided biopsy	US criterion; transverse diameter of >7 mm (level II–VI), peripheral/mixed blood flow present
Lo et al. [38]	2019	Prospective	109	46 (21–86)	109	Toshiba Aplio 500 US system (Otawara, Japan)	15-4	E_max_	66.3 ± 24.3	Cervical	US-FNA or US-CNB	-
Luo et al. [31]	2019	Prospective	118	46.7 (27–69)	121	Aixplorer ultrasound system (Supersonic Imagine, Aix-en-Provence, France)	15-4	E_max_	54.79 ± 37.42	Axillary	ALNB or SLNB	Tumour deposit > 0.2 mm in diameter in at least one lymph node
								E_mean_	49.93 ± 35.68			
								E_min_	41.88 ± 32.67			
								E_sd_	3.74 ± 3.16			
Chen et al. [34]	2018	Prospective	62	43.5 (19–66)	114	Aixplorer (SuperSonic Imagine, Aixen-Provence, France)	15-4	E_max_	31.6 (IQR: 25.2; 55.9)	Cervical	US-Guided CNB	AJCC staging system
								E_mean_	22.4 (IQR: 18.8; 36.6)			
								E_min_	15.8 (IQR: 9.6; 22.4)			
Kim et al. [30]	2018	Retrospective	43	49 (29–81)	43	Aixplorer (SuperSonic Imagine, Aix-en-Provence, France)	15-4	E_max_	50.5 (IQR: 39.9; 88.0)	Cervical	FNAB	-
								E_mean_	37.1 (IQR: 20.0; 46.3)			
								E_min_	11.3 (IQR: 4.2; 34.7)			
								E_sd_	7.8 (IQR: 4.6; 11.2)			
Seo et al. [29]	2018	Retrospective	53	54.7 (33–80) *	54	Aixplorer (Supersonic Imagine, Aix en Provence, France)	15-4	E_max_	79.80 ± 65.95	Axillary	US-guided FNAB or SLNB	US criterion
								E_mean_	55.99 ± 49.19			
								E_min_	29.29 ± 31.44			
								E_sd_	13.92 ± 11.46			
You et al. [39]	2018	Prospective	39	45.6 (15–67)	141	Aixplorer US system (SuperSonic Imagine, Aix en Provence, France)	15-4	E_max_	58.7 ± 25.7	Cervical	FNAB	US criterion 18 months follow-up
								E_mean_	30.6 ± 14.9			
								E_min_	11.9 ± 9.1			
								E_sd_	10.2 ± 5.0			
Tan et al. [40]	2017	Prospective	42	44 (23–61)	42	Aixplorer US system (SuperSonic Imagine, Aix-en-Provence, France)	SL10-2	E_max_	52.0 (IQR: 38.1; 65.1)	Neck, Supraclavicular fossze, axilla	CNB	NA
								E_mean_	16.8 (IQR: 10.6; 26.1)			
								E_min_	0.1 (IQR: 0.1; 0.4)			
								E_sd_	9.1 (IQR: 6.9; 11.7)			
Youk et al. [22]	2017	Retrospective	130	49.4 (18–84)	130	Aixplorer (SuperSonic Imagine, Aix-en-Provence, France)	15-4	E_max_	64.6 ± 41.9	Axillary	ALND and SLNB	-
								E_mean_	50.2 ± 31.8			
								E_min_	31.4 ± 24.8			
								E_sd_	9.0 ± 9.7			
Desmots et al. [41]	2016	Prospective	56	49 (25–84)	63 (62 involved in further analysis)	Aixplorer, SuperSonic Imagine, Aix-en-Provence, France) with a conventional 15- to 4-MHz transducer linear probe (SuperLinear SL15-4)	SL15–4	E_max_	72 ± 59	Head & Neck	Surgical resection, FNAC and US-follow up	AJCC staging system
Jung et al. [19]	2015	Retrospective	66	45.2	84	Aixplorer (SuperSonic Imagine, Les Jardins de la Duranne, Aix en Provence, France)	15-4	E_max_	79.61 ± 71.23	Cervical	US-Guided FNAB	US criterion
								E_mean_	67.93 ± 62.52			
								E_min_	48.49 ± 47.21			
Choi et al. [35]	2013	Prospective	15	54.2 (38–73)	67	Aixplorer (SuperSonic Imagine, Aix en Provence, France)	15-4	E_max_	41.06 ± 36.34	Cervical	Surgical resection	US criterion
Bhatia et al. [12]	2012	Prospective	46	52.8 (7–74)	55	Aixplorer; (SuperSonic Imagine, Les Jardins de la Duranne, Aix en Provence, France)	15-4	E_max_	42.2 (IQR: 28.5; 126.4)	Cervical	US-Guided FNAB	Doppler criteria
								E_mean_	25.0 kPa (IQR: 19.3; 86.2)			
Tourasse et al. [33]	2012	Prospective	65	-	81	SuperSonic Imagine device (Aix en Provence, France)	N/A	E_max_	6.71–44.18 (mean = 23.27)	Axillary	SLNB	-
								E_mean_	6.24–29.72 (mean = 17.47)			
								E_sd_	0.3–9.7 (mean = 2.95)			

* mean age calculated for cohort prior to exclusion of patients. abbreviations: CNB: core needle biopsy; ALND: axillary lymph node dissection; ALNB: axillary lymph node biopsy; SLNB: sentinel lymph node biopsy; FNAB: fine needle aspiration biopsy; AJCC: American Joint Committee on Cancer: lesion size > 2 mm, macro-metastasis; 0.2–2.0 mm, micro-metastasis; and <0.2 mm, isolated tumour cells; US: ultrasound; US criterion: at least one of the following criteria: (i) loss of hilar fat, (ii) cortical heterogeneous echogenicity, (iii) echogenic dots (calcification), (iv) a cystic or necrotic area, and (v) long- to short-axis diameter ratio < 2.0. Doppler criteria: abnormality including increased size (short axis diameter ≥ 6 mm), round shape, abnormal parenchymal echogenicity, architecture, and vascularity.

**Table 2 cancers-14-05568-t002:** Statistical characteristics of included studies.

		SWE Parameter	Cutoff Values	Number of Lymph Node Lesions	Number of Disease Positive Lymph Nodes	TP	FP	TN	FN	Sensitivity	Spec	PPV	NPV	Accuracy	AUC
Ng et al. [16]	2022	E_max_	15.2	107	50	26	25	32	24	0.52	0.56	0.51	0.57	0.54	0.61
Chami et al. [37]	2021	E_mean_	15.2	222	151	66	12	59	85	0.44	0.83	0.85	0.41	0.56	0.66 (95% CI: 0.59–0.73)
Yang et al. [32]	2021	E_max_	31.6	109	66	47	1	24	29	0.56	0.96	0.97	0.45	0.65	0.825 (95% CI 0.741–0.891)
Lo et al. [38]	2019	E_max_	42	109	24	20	30	55	4	0.83	0.65	0.40	0.93	0.69	0.688 (0.601–0.775)
Luo et al. [31]	2019	E_max_	26.05	121	60	56	7	54	4	0.93	0.89	0.89	0.93	0.91	0.94
		E_mean_	26.9	121	60	52	2	59	8	0.87	0.97	0.96	0.88	0.92	0.95
		E_min_	22.75	121	60	49	6	55	11	0.82	0.90	0.89	0.83	0.86	0.91
		E_sd_	2.05	121	60	42	5	56	18	0.70	0.92	0.89	0.76	0.81	0.83
Chen et al. [34]	2018	E_max_	20.6	114	26	26	44	44	0	1.00	0.50	0.37	1.00	0.61	0.82
		E_mean_	18.4	114	26	22	15	73	4	0.85	0.83	0.59	0.95	0.83	0.88
		E_min_	15.5	114	26	14	5	83	12	0.54	0.94	0.74	0.87	0.85	0.80
Kim et al. [30]	2018	E_max_	37.5	43	12	10	1	30	2	0.83	0.97	0.91	0.94	0.93	0.93
		E_mean_	23	43	12	8	1	30	4	0.67	0.97	0.89	0.88	0.88	0.94
		E_min_	11.7	43	12	6	4	27	6	0.50	0.87	0.60	0.82	0.77	0.70
		E_sd_	6.9	43	12	7	1	30	5	0.58	0.97	0.88	0.86	0.86	0.77
Seo et al. [29]	2018	E_max_	20.9	54	34	28	1	19	6	0.82	0.95	0.97	0.76	0.87	0.89
		E_mean_	23.8	54	34	26	0	20	8	0.76	1.00	1.00	0.71	0.85	0.88
		E_min_	11.4	54	34	21	1	19	13	0.62	0.95	0.95	0.59	0.74	0.78
		E_sd_	4.05	54	34	26	0	20	8	0.76	1.00	1.00	0.71	0.85	0.88
You et al. [39]	2018	E_max_	40.2	141	35	28	7	99	7	0.80	0.93	0.80	0.93	0.90	0.92
		E_mean_	22.1	141	35	26	12	94	9	0.74	0.89	0.68	0.91	0.85	0.87
		E_min_	12.4	141	35	19	24	82	16	0.54	0.77	0.44	0.84	0.72	0.61
		E_sd_	4.1	141	35	32	23	83	3	0.91	0.78	0.58	0.97	0.82	0.92
Tan et al. [40]	2017	E_max_	37.9	42	34	18	4	16	4	0.82	0.80	0.82	0.80	0.81	0.845 (0.701–0.938)
		E_mean_	15.5	42	34	14	0	20	8	0.64	1.00	1.00	0.71	0.81	0.732 (0.573–0.857)
		E_sd_	6.3	42	34	18	6	14	4	0.82	0.70	0.75	0.78	0.76	0.777 (0.622–0.891)
Youk et al. [22]	2017	E_max_	25.8	130	65	61	9	56	4	0.94	0.86	0.87	0.93	0.90	0.941 (0.885, 0.974)
		E_mean_	18.7	130	65	61	8	57	4	0.94	0.88	0.88	0.93	0.91	0.946 (0.892, 0.978)
		E_min_	12.3	130	65	56	8	57	9	0.86	0.88	0.88	0.86	0.87	0.915 (0.853, 0.956)
		E_sd_	4	130	65	51	7	58	14	0.78	0.89	0.88	0.81	0.84	0.900 (0.835, 0.945)
Desmots et al. [41]	2016	E_max_	31	62	30	26	4	28	4	0.87	0.88	0.87	0.88	0.87	0.903 ± 0.042
Jung et al. [19]	2015	E_max_	57	84	51	43	23	10	8	0.84	0.30	0.65	0.56	0.63	0.738 (0.633–0.843)
		E_mean_	29	84	51	39	11	22	12	0.76	0.67	0.78	0.65	0.73	0.748 (0.644–0.852)
		E_min_	24	84	51	13	0	33	38	0.25	1.00	1.00	0.46	0.55	0.737 (0.632–0.842)
Choi et al. [35]	2013	E_max_	19.44	67	34	31	1	32	3	0.91	0.97	0.97	0.91	0.94	0.96 (95% CI: 0.885, 0.993)
Bhatia et al. [12]	2012	E_max_	45	55	31	15	2	22	16	0.48	0.92	0.88	0.58	0.67	0.77 (95% CI 5 0.57–0.83)
		E_mean_	30.2	55	31	13	0	24	18	0.42	1.00	1.00	0.57	0.62	0.77 (95% CI 5 0.57–0.83)
Tourasse et al. [33]	2012	E_max_	26.4704	81	11	4	0	70	7	0.36	1.00	1.00	0.91	0.91	0.75 (95% CI: 0.55–0.95)
		E_mean_	23.5947	81	11	2	0	70	9	0.18	1.00	1.00	0.89	0.89	0.76 (95% CI: 0.58–0.94)

**Table 3 cancers-14-05568-t003:** Quality assessment of all included studies.

	QUADAS							
		Risk of Bias				Applicability Concerns		
Study	Overall Diagnostic Quality	Patient Selection	Index Test	Reference Standard	Flow and Timing	Patient Selection	Index Test	Reference Standard
Ng et al. [16]	Fair	Unclear	Low	Low	Low	Low	Low	Low
Chami et al. [37]	Good	Low	Low	Low	Low	Low	Low	Low
Yang et al. [32]	Good	Low	Low	Low	Low	Low	Low	Low
Lo et al. [38]	Good	Low	Low	Low	Low	Low	Low	Low
Luo et al. [31]	Good	Low	Low	Low	Low	Low	Low	Low
Chen et al. [34]	Good	Low	Low	Low	Low	Low	Low	Low
Kim et al. [30]	Good	Low	Low	Low	Low	Low	Low	Low
Seo et al. [29]	Good	Low	Low	Low	Low	Low	Low	Low
You et al. [39]	Fair	Low	Low	Low	Unclear	Low	Low	Low
Tan et al. [40]	Good	Low	Low	Low	Low	Low	Low	Low
Youk et al. [22]	Fair	Low	Low	Low	Unclear	Low	Low	Low
Desmots et al. [41]	Good	Low	Low	Low	Low	Low	Low	Low
Jung et al. [19]	Good	Low	Low	Low	Low	Low	Low	Low
Choi et al. [35]	Good	Low	Low	Low	Low	Low	Low	Low
Bhatia et al. [12]	Good	Low	Low	Low	Low	Low	Low	Low
Tourasse et al. [33]	Fair	Low	Low	Low	Unclear	Low	Low	Low

Colour representation: green indicates low risk of bias and yellow indicates unclear risk of bias.

## Data Availability

The data presented in this study are available on request from the corresponding author.

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
