# Peer review of "Evaluating Different Quantitative Shear Wave Parameters of Ultrasound Elastography in the Diagnosis of Lymph Node Malignancies: A Systematic Review and Meta-Analysis"

_cancers, 2022, doi:10.3390/cancers14225568_

Round 1

Reviewer 1 Report

The authors did a systematic view on the ultrasound elastography in the diagnosis of lymph node malignancies. Emax (maximum stiffness), Emean (mean stiffness), Emin (minimum stiffness), and Esd (standard deviation) were extracted for the meta-analysis of shear wave elastography. They find that SWE is a promising adjunct imaging technique in the diagnosis of lymph node malignancy. SWE parameters of Emax and Esd are sensitive in clinical screening. Overall, this paper is in a good quality. With minor revision, it can be published.

The title should emphasize ultrasound elastography.

Line 63 specify the meaning of “tissue shear”

Line 62-64 In the static elastography, the strain is detected for the imaging purpose, but it is not the force in physics. The authors should understand the meaning of all physical parameters. In addition, in the static elastography, the compressive instead of shear deformation is detected.

Line 89 elastogram equally means colour elasticity map

Line 111-112 whats the meaning of this sentence? Only an article?

Line 126 grammar error

Line 320-323 actually in the other studies, the shear modulus is also converted from the shear wave speed. Till now no approach is available to measure the shear modulus directly. Dont understand why operating at different frequency range would lead to inconsistency and inaccuracy since no dispersion is considered in all studies.

Author Response

Thank you for the comments.

Point 1: The title should emphasize ultrasound elastography.

Response 1: Thank you for this comment, we have now modified the title to “Evaluating Different Quantitative Shear Wave Parameters of Ultrasound Elastography in the Diagnosis of Lymph Node Malignancies: A Systematic Review and Meta-analysis” (line 2-3).

Point 2: Line 63 specify the meaning of “tissue shear”

Response 2: Thank you for this comment. We have now modified it to ‘tissue deformation’ and deleted ‘tissue shear’ to avoid confusion with shear waves. (line 64-66)

Point 3: Line 62-64 In the static elastography, the strain is detected for the imaging purpose, but it is not the force in physics. The authors should understand the meaning of all physical parameters. In addition, in the static elastography, the compressive instead of shear deformation is detected.

Response 3: Thank you for highlighting this. We have now expanded this sentence to give more detailed explanation. (Line 64-66)

Point 4: Line 89 “elastogram” equally means “colour elasticity map”

Response 4: Thank you. “colour elasticity map” is now deleted. (Line 91)

Point 5: Line 111-112 what’s the meaning of this sentence? Only an article?

Response 5: Thank you. It has now been corrected now. (Line 122)

Point 6: Line 126 grammar error

Response 6: Thank you. This has now been corrected. (Line 138)

Point 7: Line 320-323 actually in the other studies, the shear modulus is also converted from the shear wave speed. Till now no approach is available to measure the shear modulus directly. Don’t understand why operating at different frequency range would lead to inconsistency and inaccuracy since no dispersion is considered in all studies.

Response 7: Thank you for highlighting this. We were trying to explain the rationale of not including studies that reported SWS values even though the conversion is available. The EFSUMB Guidelines 2017 mentioned that ‘SWS will vary with shear wave frequency’ and have ‘frequency dependencies’ and so we believe if the linear array transducer operated at a different bandwidth (4-9 MHz) rather than 4-15 MHz, it could introduce greater inconsistencies when converting the values to elastic modulus. Also, converting SWS values to kPa values manually would mean that the means and standard deviations will not be equal due to the squared relationship. And thus, we decided to not include those studies reporting results in SWS and not to convert the results ourselves. We have modified the text and quoted the updated EFSUMB guideline to explain this better. (Line 344-352)

EFSUMB Guidelines 2017:

Dietrich, C.F.; Bamber, J.; Berzigotti, A.; Bota, S.; Cantisani, V.; Castera, L.; Cosgrove, D.; Ferraioli, G.; Friedrich-Rust, M.; Gilja, O.H.; et al. EFSUMB Guidelines and Recommendations on the Clinical Use of Liver Ultrasound Elastography, Update 2017 (Long Version). Ultraschall Med 2017, 38, e16-e47, doi:10.1055/s-0043-103952.

Reviewer 2 Report

The paper is very interesting and could improve the clinical practice. 

In study characteristics the authors says that the studies included were between 2102 and 2022, but in the tables 1 and 2 the studies appear between 2012 and 2021. 

With the exception of one study (performed on Toshiba ultrasound scanners) considered, all articles included in this meta-analysis are performed on Aixplorer - Supersonic equipment. However, the data obtained are inhomogeneous ... Do the authors have experience with SWE on ultrasound machines from Samsung or Philips for example? 

The 2013 EFSUMB Guidelines for the application of soft tissue elastography were mentioned and cited. There is a newer guideline with EFSUMB recommendations published in 2108. 

Author Response

Thank you for the comments.

Point 1: In study characteristics the authors says that the studies included were between 2012 and 2022, but in the tables 1 and 2 the studies appear between 2012 and 2021. 

Response 1: Thank you for spotting this error. Study by Ng et al. appears was initially published online in April 2021 and later in print in Jan 2022. We have now updated our reference list and quoted the correct year of publication.

Point 2: With the exception of one study (performed on Toshiba ultrasound scanners) considered, all articles included in this meta-analysis are performed on Aixplorer - Supersonic equipment. However, the data obtained are inhomogeneous ... Do the authors have experience with SWE on ultrasound machines from Samsung or Philips for example? 

Response 2: Thank you for this comment. We don’t have experience with SWE on ultrasound machines from Samsung or Philips ourselves however, the use of Toshiba, Philips and Mindray in 2D-SWE have been recognised as recent developments, updated in EFSUMB 2017 guideline in comparison with EFSUMB 2013 when only Supersonic and Siemens were useable for 2D-SWE. The updated guideline described these manufacturers as broadly similar at offering 2D-SWE but differ with respect to the details of the method and sampling rate. Since the Supersonic technique has been available for quite some time, we think this explains why most of the included studies have implemented this technique. We recognize that there could be differences between vendors especially when they are continually developing new techniques and softwares. We have mentioned this in our discussion as a potential future research question (line 424-431).

Point 3: The 2013 EFSUMB Guidelines for the application of soft tissue elastography were mentioned and cited. There is a newer guideline with EFSUMB recommendations published in 2018. 

Response 3: Thank you for this comment. Yes, the 2013 guidelines were mainly cited as the physics and principles of the elastography technology were better explained in this earlier guideline; the 2017 guidelines, although were focusing on the use of ultrasound elastography in liver, the principles and updated machinery developments were better explained there in details. The 2018 guideline is an updated version that expanded on the use of US elastography on different non-hepatic organs. We have updated our references list and included the newest guidelines in 2018 within our text where appropriate. 

Reviewer 3 Report

This is a systemic review and meta-analysis for evaluating different quantitative Shear Wave Elastography parameters in the diagnosis of lymph node malignancies. The authors searched Medline, Ovid, PubMed, Embase and Web of Science databases up to date August 2022 for studies that reported on different quantitative SWE parameters with different cut-off values for the diagnosis of lymph node malignancies. Study quality was assessed using the QUADAS-2 tool. Through to certain inclusion and exclusion process, the authors at last included 16 studies (1276 participants with 1479 lymph nodes) for final analysis. Results showed Emax and Esd demonstrated highest pooled sensitivity (0.78 (95% CI: 0.690.87); 0.78 (95% CI: 0.680.87. Since differentiating malignant lymph nodules from begin lymph nodes is very important in clinic practice, the potential readers might be very interested in this study.

Strengths

1. Reasonable methodology and appropriate outcome presentation.

2. Well written.

Weakness

High heterogeneity among included studies.

Comments in details:

1.       The heterogeneity of included studies should be tested or evaluated by Chi square test or I2 statistic in Materials and Methods section.

2.       P7 Line 176-179: the accuracy in the diagnosis of LN malignancy is different among FNAB, CNB and surgically resection.

3.       P7 Line 182-185: there are huge discrepancies in cut-off values of different SWE parameters among different studies. Is it rational to do data synthesis for studies with high heterogeneity?

4.       P12 Line 272: “0.97” should be “0.87”

5.       P13 Line 279-280: I2 statistic results should be moved to Materials and Methods section.

6.       P13 line 283: should “in this study” be “in these studies”?

Author Response

Thank you for the comments.

Point 1: The heterogeneity of included studies should be tested or evaluated by Chi square test or Istatistic in Materials and Methods section.

Response 1: Thank you for this comment. We have now added a Meta-analysis section in Materials and Methods, to explain our methodology more thoroughly including the I2 heterogeneity evaluation. (Line 184-191)

Point 2: P7 Line 176-179: the accuracy in the diagnosis of LN malignancy is different among FNAB, CNB and surgically resection.

Response 2: Thank you for highlighting this. We are aware of the potential impact of using different gold standards in terms of their intrinsic diagnostic accuracies and we recognize this as a study limitation contributing to the reported high heterogeneities. We also understand the difficulties in practice when choosing biopsy methods during clinical trials and it is inevitable that different gold standards might be used. We believe that it would still be beneficial to describe the broad spectrum of the overall diagnostic accuracies of the SWE technology in LN malignancies, while recognizing that the difference in gold standards could have contributed to heterogeneities, and the implication of this is that future trials should take care of comparing the type of gold standard test used which we acknowledged in our discussion. (Line 409-417)

Point 3: P7 Line 182-185: there are huge discrepancies in cut-off values of different SWE parameters among different studies. Is it rational to do data synthesis for studies with high heterogeneity?

Response 3: Thank you for this comment. This is one of the main limitations of this meta analyses. However, we believe an up to date quantitative assessment and account of developments on this topic is needed to encourage other research groups to appraise the limitations identified across various studies then find ways to mitigate them in subsequent experiments or trials. This is not uncommon for new technologies that are undergoing the translational jump from lab to clinic.

Point 4: P12 Line 272: “0.97” should be “0.87”

Response 4: Thank you. It has now been corrected. (line 295)

Point 5: P13 Line 279-280: Istatistic results should be moved to Materials and Methods section.

Response 5: We have now added a sub-section called meta-analysis in Material and Methods to explain our statistical methods thoroughly. (line 184-191)

Point 6: P13 line 283: should “in this study” be “in these studies”?

Response 6: Thank you. This has now been corrected. (line 307)

Round 2

Reviewer 3 Report

In my opinion, although the topic discussed in the manuscript is very interesting, due to the high heterogeneity(I2 >80%)among included studies, it is inappropriate to do such a meta-analysis. The manuscript  does not provide new information.

Author Response

Response to Reviewer Comments

In my opinion, although the topic discussed in the manuscript is very interesting, due to the high heterogeneity(I2 >80%)among included studies, it is inappropriate to do such a meta-analysis. The manuscript does not provide new information

Response:

Dear reviewer, thank you for pointing this out. High heterogeneity is one of the main limitations of this meta-analysis. However, we wish to attempt exploring the possible causes of high heterogeneity. We have postulated some potential causes of this limitation, starting from line 405, and we have commented variabilities across studies including different gold standards used. Hopefully, this provides new insights to future research groups to find ways to mitigate them in subsequent trials. Finally, we have pointed out that our results should be interpreted with caution with the high heterogeneity (lines 469-473), in order to be transparent with our discussion and conclusion. Our approach to the discussion of heterogeneity is in line with the Cochrane handbook [1].

Moreover, high heterogeneity is not uncommon in developing technologies that are undergoing the translational jump from lab to clinic. We have noticed that in the existing published meta-analysis also exhibited high heterogeneity [2,3], they have address it by stratifying their meta-analyses and given explanations for the cause of high heterogenies. Therefore, we have adopted a similar approach in our review and we believe this meta-analysis will contribute to the growing body of knowledge in this field.  

Best regards,

Yujia Gao

  1. Higgins JPT, Thomas J, Chandler J, Cumpston M, Li T, Page MJ, Welch VA (editors). Cochrane Handbook for Systematic Reviews of Interventionsversion 6.3 (updated February 2022). Cochrane, 2022. Available from www.training.cochrane.org/handbook.
  2. Huang, X.-w.; Huang, Q.-x.; Huang, H.; Cheng, M.-q.; Tong, W.-j.; Xian, M.-f.; Liang, J.-y.; Wang, W. Diagnostic Performance of Quantitative and Qualitative Elastography for Axillary Lymph Node Metastasis in Breast Cancer: A Systematic Review and Meta-Analysis. Front Oncol 2020, 10, doi:10.3389/fonc.2020.552177.
  3. Wang, R.Y.; Zhang, Y.W.; Gao, Z.M.; Wang, X.M. Role of sonoelastography in assessment of axillary lymph nodes in breast cancer: a systematic review and meta-analysis. Clin Radiol 2020, 75, 320.e321-320.e327, doi:10.1016/j.crad.2019.11.016.
